# Recurrence of Nephrolithiasis and Surgical Events Are Associated with Chronic Kidney Disease in Adult Patients

**DOI:** 10.3390/medicina58030420

**Published:** 2022-03-12

**Authors:** Martha Medina-Escobedo, Katy Sánchez-Pozos, Ana Ligia Gutiérrez-Solis, Azalia Avila-Nava, Lizeth González-Rocha, Roberto Lugo

**Affiliations:** 1Research Unit, Hospital Regional de Alta Especialidad de la Peninsula de Yucatán, Calle 7 por 20 y 22, Fraccionamiento Altabrisa, Merida 97130, Mexico; marthamedinaescobedo@hotmail.com (M.M.-E.); ana.gutierrez@salud.gob.mx (A.L.G.-S.); azalia.avila@salud.gob.mx (A.A.-N.); lizglezrocha@gmail.com (L.G.-R.); 2Molecular Endocrinology Laboratory, Hospital Juarez de Mexico, Av. Instituto Politecnico Nacional 5160, Gustavo A. Madero, Mexico City 07760, Mexico; katypozos@gmail.com

**Keywords:** renal lithiasis, endemic disease, risk factors, kidney injury

## Abstract

*Background and objective*: Nephrolithiasis (NL) is a public health problem in the population of Southeast Mexico because of its high prevalence and recurrence. The evolution of this pathology can result in renal damage and may even cause chronic kidney disease (CKD), leading to a reduced glomerular filtration rate (GFR), decreased kidney function, and kidney loss in advanced stages. However, few studies support this evidence in the population. The present study aimed to determine risk factors associated with CKD in adult patients in an endemic population of Mexico. *Materials and methods*: A case-control study was carried out with patients diagnosed with NL. Additionally, the clinical information of patients (age, weight, height, blood pressure, comorbidities, and time of progress of NL), characteristics of the stones (number, location, and Hounsfield units), and biochemical parameters were collected. *Results*: The recurrence of NL was associated with CKD (OR 1.91; 95% CI 1.37–2.27; *p* = 0.003). In addition, male sex (*p* = 0.016), surgical history (*p* = 0.011), bilateral kidney stones (*p* < 0.001), and urinary tract infections (*p* = 0.004) were other factors associated with CKD. Interestingly, thirty-two patients younger than 50 years old with >2 surgical events presented a significant decrease in GFR (*p* < 0.001). *Conclusions*: The recurrence of NL and the number of surgical events were risk factors associated with CKD in patients with NL treated in our population.

## 1. Introduction

Nephrolithiasis (NL) is a disease characterized by a saturation of crystals in the nephrons that can generate kidney stones and damage renal tissue and function [1]. Genetic and metabolic factors affect the probability of this saturation, which is high in people with metabolic disorders such as gout, renal tubular acidosis, and hypercalciuria [2]. Most NL cases are diagnosed incidentally in the initial phases, and they remain asymptomatic in follow-up examinations at 3–5 years, or until partial or total obstruction of a segment of the urinary tract occurs. When this latter occurs, the main symptom is colic pain in the lumbar fossa or some radiation; haematuria is the second most frequent symptom reported [3].

The worldwide prevalence of NL ranges from 1% to 20%, varying according to age, sex, race, geographic location, and environmental factors. For example, a higher incidence has been reported in men 40–60 years old compared with women [4]. In Mexico, the prevalence is estimated to be 2.4 cases per 100,000 inhabitants. Specifically, the state of Yucatan reports 5.8 cases per 100,000 inhabitants, the highest prevalence reported in the country; therefore, it has come to be considered an endemic area for this pathology [5].

Patients with NL have a 51–68% chance of developing chronic kidney disease (CKD) [6]. Chronic kidney disease is defined as the presence of abnormalities of kidney structures that reduce the glomerular filtration rate (GFR) and/or the presence of renal damage [7,8].

The Centers for Disease Control and Prevention estimated that 15% of the adult population (37 million people) in the USA presented CKD in 2019, being more common in people aged 65 and older (38%) and less prevalent in people 54–65 (13%) and 18–44 (7%) [9] years old. In Mexico, the incidence of CKD patients is 377 cases per million inhabitants, and the prevalence is 1142 per million inhabitants [10].

On the other hand, there is an increased prevalence of non-communicable diseases (NCDs) in the Mexican population. According to the National Health and Nutrition Survey, 2018(ENSANUT), type-2 diabetes mellitus (T2DM) is present in 15.3%, hypertension represents 12.9%, and obesity is prevalent in 35.3% of the population over 20 years old. In the same survey, Southeast Mexico had a prevalence of 15.4% for T2DM, 32.7% for hypertension, and 36.1% for obesity [11]. In this context, some studies have described that T2DM, hypertension, and obesity are risk factors for NL, and in the long term, they increase the probability of developing CKD [12]. Additionally, the Hispanic population has a 2-fold likelihood of developing CKD compared with the Caucasian population [13].

Due to the lack of evidence of risk factors associated with CKD in patients with NL in the Mexican population and the high prevalence and recurrence of this pathology, we conducted the present study to identify the risk factors associated with CKD in an endemic population Mexico.

## 2. Materials and Methods

The study design was case-control and cross-sectional. Patients over 18 years old with NL diagnoses verified by computed tomography scan and treated at the Urology Department of the Regional High Specialty Hospital of the Yucatan Peninsula (HRAEPY in the Spanish acronym) between January 2018 and June 2020 were included. Patients with acute kidney injury (AKI) and prostatic hypertrophy diagnoses were excluded. 

### 2.1. Selection of the Study Participants

The sample size was calculated using OpenEpi software (OpenEpi has been developed at the Rollins School of Public Health at Emory University, Atlanta, GA, USA) for an unmatched case-control study (95% confidence level, 5% margin error). Subjects were divided into two groups according to their GFR values. The control group consisted of patients with NL without CKD (GFR ≥ 60 mL/min/1.73 m^2^). The case group comprised patients with NL and CKD (GFR < 60 mL/min/1.73 m^2^). In both groups, GFR was calculated using the CKD-EPI formula. 

For each patient, the following information was recorded: age, sex, weight, height, arterial pressure, history of NCDs, evolution time of NL expressed in months (time with NL since diagnosis until surgical procedure), recurrence of NL, presence of CKD, history of recurrent urinary infection, and general urine test and urine culture results. Additionally, the nutritional status of the patients was assessed using the body mass index (BMI).

According to the surgical procedures, all patients who underwent kidney stone extraction at the host hospital underwent endourological procedures. Thus, the main surgical procedures performed were percutaneous nephrolithotomy and laparoscopic nephrectomy in patients with irreversible kidney damage. Nephrostomy and abscess drainage were carried out to improve kidney function. The hospital also received patients from other hospitals with a diagnosis of NL and a history of previous surgeries. The urology department evaluated these patients, and the previous diagnosis was confirmed. They were scheduled for surgical intervention when required. From all cases, surgical history was obtained (patients with previous surgical procedures) for NL and the number and type of surgical procedures.

Radiological characteristics of the stones, such as the number, size (length and width), location (right kidney, left kidney, or bilateral), radiodensity (expressed in Hounsfield units), and the presence of staghorn stones, were documented. In addition, biochemical parameters (haemoglobin, urea, serum creatinine, and serum uric acid) were also recorded.

### 2.2. Statistical Analyses

Continuous variables were evaluated using the Kolmogorov–Smirnov test to analyse the type of distribution. Student’s *t*-test or Mann–Whitney U-test were performed to compare the mean or median. Data were expressed as the mean ± standard deviation or median and quartiles. Frequencies were expressed in percentages. Differences between proportions were analysed using the Mantel–Haenszel chi-square test. Correlations between two variables were evaluated using the Pearson test. Multivariable logistic regression tests were used to explore NL risk factors. The variables were calculated using Jamovi software, computer software version 1.6, (Jamovi project 2021, Sydney, Australia). Values of *p* < 0.05 were considered statistically significant.

## 3. Results

In this study, a total of 204 patients with NL were included: 79 patients with CKD (45 women, 57.0%; and 34 men, 43.0%) and 125 patients without CKD (94 women, 75.2%; and 31 men, 24.8%). Figure 1 shows the general characteristics of patients with NL according to CKD.

The analysis (Figure 1) revealed a statistical significance for age (*p* = 0.001) and systolic blood pressure (*p* = 0.004) when patients with and without CKD were compared. As expected, haemoglobin, uric acid, urea, serum creatinine, and GFR showed statistically significant differences when patients with and without CKD were compared (*p* < 0.001) (Table 1). Only one patient showed xanthogranulomatous pyelonephritis.

No significant differences were observed when comparing stone size (Mann–Whitney U-Test, *p* = 0.742), the number of stones (Mann–Whitney U-Test, *p* = 0.742), presence of staghorn stones (Mantel–Haenszel Chi-squared test, *p* = 0.960), and Hounsfield units (HU) (Mann–Whitney U-Test, *p* = 0.057) between patients with and without CKD (Table 2). In addition, results indicated that the main composition of the kidney stone in the same population was calcium oxalate monohydrate, which corresponds with their HU located in the case and control groups (~900 HU) [14,15]. 

Table 3 shows the family, personal, and surgical history of the patients included in this study according to the presence or absence of CKD. Male sex, hypertension, urine infection, recurrence of NL, and surgical history were associated with CKD. The association remained for all clinical characteristics after adjusting for BMI, sex, and age, except for hypertension. Extracorporeal lithotripsy was performed in eight patients with CKD and five without CKD (*p* = 0.138). Additionally, T2DM was statistically significant in the adjusted analysis.

Regarding the location of the kidney stones, the right kidney (Mantel–Haenszel Chi-squared test, OR 0.5; 95% CI 0.26–1.02; *p* = 0.078) and left kidney (Mantel–Haenszel Chi-squared test, OR 0.5; 95% CI 0.27–1.12; *p* = 0.138) did not show significant differences between patients with and without CKD. Only having stones in both kidneys (bilateral) was statistically significant in patients with CKD compared with patients without CKD (Mantel–Haenszel Chi-squared test, OR 3.2; 95% CI 1.75–5.83; *p* < 0.001).

As age and invasive surgical history were statistically significant when compared between groups, a comparative analysis according to GFR was proposed. We divided the patients by age (≤50 and >50 years old) to control the impact of age on GFR and by the number of surgical events (≤2 and >2 surgical events). These results are expressed in Table 4.

Patients 50 years old or younger had more surgical events than older patients, which resulted in a decrease in GFR values and more cases of CKD (Table 4; Figure 2). In patients >50 years old, GFR was analysed and showed no significant differences between groups in relation to the number of surgical events (*p* = 0.954) and frequency of CKD (*p* = 0.783).

A multivariable logistic regression of CKD patients was performed to determine whether the recurrence of NL is a risk factor for CKD, independent of hypertension and urinary tract infections. The results showed that recurrence of NL remained statistically significant independent of hypertension (OR 1.1; 95% CI 1.05–2.83; *p* = 0.037) and urinary tract infections (OR 1.3; 95% CI 1.04–3.22; *p* = 0.044). However, when hypertension and urinary tract infections were considered in conjunction, the significance was lost (OR 1.2; 95% CI 0.97–3.24; *p* = 0.164).

## 4. Discussion

Currently, NL is considered a public health problem in the Yucatan Peninsula, according to the Statistics of the Health Services of Yucatan State in Southeast Mexico [16]. In addition, CKD is one of the main complications in patients with NL [12].

The host hospital in the present study is a reference centre for patients with complex urolithiasis-related problems that eventually will require surgical intervention for NL. Moreover, patients usually show complications or present comorbidities (T2DM, hypertension, and prostatic hypertrophy, among others), thus promoting renal impairment.

Although women were the majority in this study, the male sex was a risk factor in patients with CKD, also representing the older population. In contrast, a study by Shoag et al., found that a history of kidney stones was associated with a statistically significant increased risk of CKD among women. Nonetheless, no significant association was evident in men. Shoag et al., proposed a characteristic pathophysiology process in each gender [17]. While in the present study, the men were at higher risk than women, the findings suggest a different disease progress in each gender.

Unadjusted results indicated that hypertension is a risk factor associated with CKD, but the significance was lost when the results were adjusted, although it should be noted that there was still a trend, which could be due to the number of participants. In this context, many studies have confirmed the role of hypertension in the development of CKD thus, we do not rule out the impact of hypertension, and this effect was attributed to the sample size [18].

The increase in prevalence of CKD in our population may be due to the coexistence of several clinical associations that accelerate renal function damage. Although T2DM was a statistically significant factor in the adjusted analysis by BMI, sex, and age, it was more frequent in patients without CKD, suggesting that T2DM is not an exclusive risk factor for CKD. Moreover, it is important to mention that factors such as the disease course and age of diagnosis were not considered. In this regard, studies suggest that T2DM and factors such as NL, urinary infections, and obesity can increase the risk of irreversible kidney damage, but further studies with an increased sample size that include data on disease course and age of diagnosis are needed [19,20]. 

Previous studies have shown an association between staghorn stones and CKD [19]. Interestingly, this study did not show a significant difference between groups in staghorn stones frequency, which suggests that recurrence of NL and surgical events are independent risk factors for CKD. Remarkably, the stone sizes and composition (HU) in both groups were similar.

Urinary tract infections were risk factors for CKD in female patients (OR 0.36; 95% CI 0.14–0.93; *p* = 0.032). Infectious processes are usually frequent in patients with NL hence, these infections could be due to an obstructive process that can damage the kidney parenchyma, leading to increased morbidity and a high risk of mortality [21]. Furthermore, urinary tract infections due to acute pyelonephritis can lead to chronic scarring kidney injury, characterized by tubulointerstitial involvement, and manifested by polyuria, sodium loss, and hypertension [22].

Recurrence of NL was associated with CKD. The recurrence of NL is related to the consumption of hard water in the Yucatan Peninsula (>400 ppm) [23]. In addition, the high consumption of carbonated beverages and diets rich in sodium, calcium, and animal protein can cause the formation of new stones in the urinary tract [24,25]. Unfortunately, we could not evaluate the diet and stone composition of the patients included in this study; however, it is well known that the population of the Yucatan Peninsula showed the highest NCDs scores of all populations in Mexico, according to the ENSANUT 2018 survey [11]. Therefore, it is advisable to individualize cases and analyse the composition of the stones to provide adequate treatment to patients and thus avoid future stone recurrences as much as possible. In addition, further studies that consider nutrition or drug interventions in association with the recurrence of NL are needed to observe whether these strategies can correct metabolic alterations or control urinary tract infections, which are the principal leading causes for the progression of NL to kidney damage.

As expected, the results showed that patients with NL with CKD were older (*p* = 0.001), and age was a risk factor. Likewise, the percentage of patients with an invasive surgical history also significantly differed between those with and without CKD (43% vs. 28.3%, *p* < 0.001), suggesting a relationship between both parameters. Glassock et al., in their reports, suggested that GFR decreases with age, even in healthy older patients [26]. To analyse this relationship, patients were grouped according to age (≤50 years old and >50 years old) and the number of surgical events. A decrease in GFR was observed according to the increase in the number of surgical interventions, showing a strong correlation between both parameters (Pearson correlation coefficient, r = −0.91). This decrease in GFR suggests that the kidney cannot restore its normal function after each surgical event or even achieve adequate function. In other words, when the recurrence of NL and the elimination of the subsequent stone through surgical procedures are performed, the damage to kidney function increases significantly, and GFR decreases. The CKD frequency in patients aged ≤50 years old and >2 surgical events were significantly higher compared to those who reported fewer surgical events (*p* < 0.001) (Table 4). In support of these data, some authors have reported the impact of long-term surgical events on kidney function [19].

On the other hand, no statistically significant differences were observed when comparing patients older than 50 years with more than two surgical events versus patients younger than 50 years old. Therefore, the high risk conferred by number of surgical events was independent of age. It is important to mention that significance remained when analysing whether recurrence of NL was a risk factor for CKD independent of hypertension and urinary tract infections.

The study showed some biases (Berkson bias): (a) all patients were selected from the High Specialty Hospital, who were referred from other medical units; (b) most of them had comorbidities (T2DM, obesity, hypertension); (c) as the present study was cross-sectional, it was not possible to know the GFR of patients prior to each surgical event. In addition, some of them lived in a rural area without enough resources to cover the cost of traveling and attending all medical appointments, leading to an incorrect medical control of the disease progression.

Regarding urine supersaturation, this was not evaluated because the excretion of urinary components are modified once the GFR is impaired. Hence, we considered that it was not appropriate to include this comparison with patients with CKD.

## 5. Conclusions

The recurrence of NL and the number of surgical events were risk factors associated with CKD in patients with NL in our population. Additionally, male sex, bilateral kidney stones, and urinary tract infections were also risk factors for CKD in patients with NL. These findings indicated that independent of the type of surgical procedure, recurrence of NL increases the risk of irreversible kidney damage, and many patients may lose a kidney and require substitutive therapy, which increases mortality in the population.

## Figures and Tables

**Figure 1 medicina-58-00420-f001:**
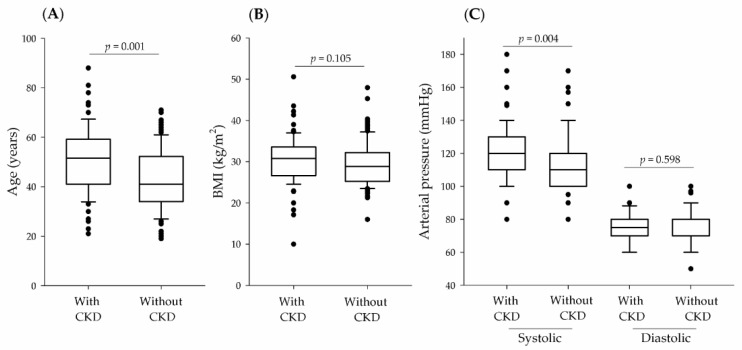
General characteristics of patients with NL according to GFR classification (presence, or absence of CKD). (**A**), Age; (**B**), Body mass index (BMI); (**C**), Systolic and diastolic arterial pressure.

**Figure 2 medicina-58-00420-f002:**
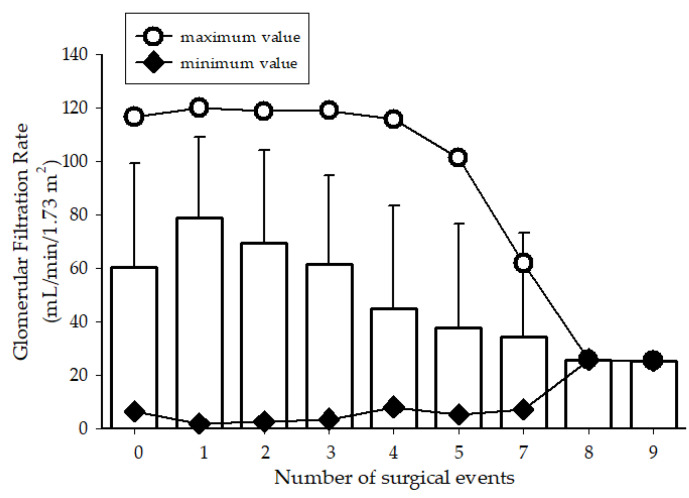
Glomerular filtration rate vs. the number of surgical events in patients ≤ 50 years old with nephrolithiasis.

**Table 1 medicina-58-00420-t001:** Clinicals characteristics of patients with NL.

Characteristics	with CKD 79 (41.1)	without CKD * 113 (58.9)	*p*-Value
Haemoglobin (mmol/L)	7.01 (6.02–7.63)	7.63 (6.83–8.5)	<0.001
Uric acid (mmol/L)	0.35 (0.26–0.43)	0.26 (0.21–0.29)	<0.001
Urea (mmol/L)	8.32 (5.70–12.69)	4.02 (3.37–4.99)	<0.001
Serum creatinine (umol/L)	179.49 (134.4–296.21)	69.85 (57.47–81.35)	<0.001
GFR (mL/min/1.73 m^2^)	29.9 (15.5–46.1)	98.1 (78.1–112.5)	<0.001

The parameter was analysed by Mann–Whitney U-test expressed by median and quartile. GFR: Glomerular filtration ratio. * Patients with hyperfiltration were not included.

**Table 2 medicina-58-00420-t002:** Characteristics of the kidney stones.

Characteristics	with CKD 79 (41.1)	without CKD * 113 (58.9)	*p*-Value
Number of stones ^†^	2 (1–12)	2 (1–14)	0.740
Stone sizes (mm)	30.0 (15.0–51.0)	29.0 (18.0–48.0)	0.742
HU	900 (500–1117)	990 (773–1200)	0.057
Staghorn stones ^‡^	32 (40.5)	44 (38.9)	0.960
Stone location:			
Right kidney	10 (12.6)	34 (30.0)	0.008
Left kidney	13 (16.4)	29 (25.6)	0.180
Bilateral kidney	21 (26.5)	8 (7.0)	<0.001
Multiple	35 (44.5)	42 (37.4)	0.399

The parameter was analysed using Mann–Whitney U-test expressed by median and quartile, and Mantel–Haenszel Chi-squared test. HU: Hounsfield units. * Patients with hyperfiltration were not included. ^†^ Median (minimum and maximum value). ^‡^ Staghorn stone were considered with >50 mm of the stone.

**Table 3 medicina-58-00420-t003:** Comparative analysis of family, personal, and surgical history according to its presence or absence of CKD in patients with nephrolithiasis.

			without Adjusted		Adjusted by BMI, Sex, and Age	
Characteristics, *n* (%)	with CKD 79 (41.1)	without CKD * 113 (58.9)	OR (95% CI)	*p*-Value	OR (95% CI)	*p*-Value
Family history of NL	8 (10.1)	16 (14.1)	0.68 (0.27–1.68)	0.542	1.04 (0.40–2.71)	0.929
Male sex ^†^	32 (40.5)	26 (23.0)	2.62 (1.35–4.72)	0.015	2.21 (1.25–4.23)	0.016
Obesity ^‡^	42 (53.2)	67 (59.3)	0.77 (0.43–1.39)	0.487	0.73 (0.46–1.37)	0.329
Hypertension	27 (34.2)	22 (19.5)	2.14 (1.11–4.14)	0.021	1.30 (0.89–2.23)	0.048
T2DM	11 (13.9)	29 (25.7)	0.46 (0.21–1.01)	0.073	0.28 (0.13–0.68)	0.040
Urinary tract infections	34 (43.0)	30 (26.5)	2.09 (1.13–3.84)	0.017	2.72 (1.49–5.79)	0.004
Recurrence of NL	10 (12.3)	15 (13.3)	0.94 (0.40–2.23)	0.052	1.91 (1.37–2.27)	0.003
Renal exclusion	1 (1.3)	5 (4.0)	0.27 (0.03–2.41)	0.416	0.08 (0.69–1.09)	0.998
Surgical history ^§^	34 (43.0)	32 (28.3)	1.91 (1.04–3.50)	0.045	1.80 (1.23–3.02)	0.011

The Mantel–Haenszel Chi-square test was used for data analysis. CKD: chronic kidney disease; NL: nephrolithiasis; T2DM: Type 2 diabetes mellitus. * Patients with hyperfiltration were not included. ^†^ Adjusted by BMI and age. ^‡^ Adjusted by sex and age. ^§^ Percutaneous nephrolithotomy, nephrostomy, abscess drainage, laparoscopic nephrectomy.

**Table 4 medicina-58-00420-t004:** Comparative analysis of the GFR and CKD frequency between the invasive surgical events and the age in patients with NL.

Characteristic, *n* (%)	Age ≤ 50 Years Old 128 (62.7)			Age > 50 Years Old 76 (37.3)		
	≤2 Surgical Events 96 (75.0)	>2 Surgical Events 32 (25.0)	*p*-Value	≤2 Surgical Events 57 (75.0)	>2 Surgical Events 19 (25.0)	*p*-Value
GFR (mL/min/1.73 m^2^)	75.00(45.95–100.90)	48.30 (17.80–85.60)	<0.001	57 ± 29.56	54.17 ± 25.72	0.954
CKD	17 (17.7)	20 (62.5)	<0.001	32 (56.1)	10 (52.6)	0.783

The parameter was analysed by Mann–Whitney U-test expressed by median and quartile, and by mean ± standard deviation using Student’s *t*-test. The CKD frequency was analysed using the Mantel–Haenszel Chi-square test. GFR: Glomerular filtration rate; CKD: chronic kidney disease.

## Data Availability

Data underlying this work are available upon reasonable request. Requests for data should be addressed to the corresponding author.

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
