# Peer review of "Recurrence of Nephrolithiasis and Surgical Events Are Associated with Chronic Kidney Disease in Adult Patients"

_medicina, 2022, doi:10.3390/medicina58030420_

Round 1

Reviewer 1 Report

 Results showed that recurrence of NL (OR 1.91; 95%CI 1.37–2.27; p=0.003) was a risk factor for developing associated with CKD (you have not proven causation, it's only an observation/association). In addition, other risk factors associated with CKD were male sex (p=0.016), surgical history (p=0.011), 22 bilateral kidney stones (p<0.001), and urinary tract infections (p=0.004). Interestingly, twenty-three patients 23 younger than 50 years old with >2 surgical events presented a significant decrease in GFR (p<0.001), 

Please re-write your results and discussion to reflect the association and not causation 

Author Response

Thank you for taking the time to review this manuscript and provide us your comments and suggestions. Please, find down below the answer to your comments and suggestions point by point.

Results showed that recurrence of NL (OR 1.91; 95%CI 1.37–2.27; p=0.003) was a risk factor for developing associated with CKD (you have not proven causation, it's only an observation/association). In addition, other risk factors associated with CKD were male sex (p=0.016), surgical history (p=0.011), 22 bilateral kidney stones (p<0.001), and urinary tract infections (p=0.004). Interestingly, twenty-three patients 23 younger than 50 years old with >2 surgical events presented a significant decrease in GFR (p<0.001), 

Please re-write your results and discussion to reflect the association and not causation 

Response: The manuscript was modified according to your recommendations. The results and discussion were re-writing, reflecting the association of the risk factors with CKD.

Reviewer 2 Report

In this study the authors compared risk factors between two groups of people with and without chronic kidney disease (CKD). They concluded that patients who had more previous interventions are at higher risk of developing CKD: “This decrease in GFR suggests that the kidney cannot restore its normal function after each surgical event or even achieve adequate function. In other words, when the recurrence of NL and the elimination of the subsequent stone through surgical procedures are performed, the damage to kidney function increases significantly, and GFR decreases.”

 Unfortunately, methodology has biased this result of the study. In fact, if we want to evaluate the effect of the intervention on CKD in a group of patients with urolithiasis, we should compare the rate of CKD in patients with similar stone characteristics who underwent or did not underwent surgery. As we know staghorn stones are associated with a high risk of CKD and even death if left untreated and it is predictable that if these patients left untreated, they are at high risk of CKD. Meanwhile, even if we do intervention in these patients, the CKD outcome will be worse compared with those who had no history of urolithiasis or those who have small stones or less recurrent urolithiasis that requires less intervention. In fact, the CKD is related to the complexity of the stone itself that requires intervention not a consequence of the intervention. Therefore, due to a significant bias due to basic methodological flaw.

There are other issues that all are related to the methodology. For example, as shown in table 1, patients were not matched regarding age and hypertension. Renal function status before intervention is not clear. If we want to relay on the finding, we should conclude that type2DM is associated with lower risk of CKD (table 2) in patients with kidney stone!

Author Response

Thank you for taking the time to review this manuscript and provide us with your comments and suggestions. Please, find down below the answer to your comments and suggestions point by point.

In this study, the authors compared risk factors between two groups of people with and without chronic kidney disease (CKD). They concluded that patients who had more previous interventions are at higher risk of developing CKD: “This decrease in GFR suggests that the kidney cannot restore its normal function after each surgical event or even achieve adequate function. In other words, when the recurrence of NL and the elimination of the subsequent stone through surgical procedures are performed, the damage to kidney function increases significantly, and GFR decreases.”

Unfortunately, methodology has biased this result of the study. In fact, if we want to evaluate the effect of the intervention on CKD in a group of patients with urolithiasis, we should compare the rate of CKD in patients with similar stone characteristics who underwent or did not underwent surgery.

Response 1: Regarding to the bias in the study, we added a paragraph with the study limitations.

“The study showed some biases (Berkson bias): a) All patients were selected from the High Specialty Hospital, who were referred from other medical units; b) Most of them had comorbidities (T2DM, obesity, hypertension); c) Due to the present study was cross-sectional, it was not possible to know the GFR of patients, previous to each surgical event. In addition, some of them lived in a rural area without enough resources to cover the cost of traveling and attending all medical appointments, leading to an incorrect medical control of the disease progression”.

As we know staghorn stones are associated with a high risk of CKD and even death if left untreated and it is predictable that if these patients left untreated, they are at high risk of CKD. Meanwhile, even if we do intervention in these patients, the CKD outcome will be worse compared with those who had no history of urolithiasis or those who have small stones or less recurrent urolithiasis that requires less intervention. In fact, the CKD is related to the complexity of the stone itself that requires intervention not a consequence of the intervention. Therefore, due to a significant bias due to basic methodological flaw.

Response 2: Thank you for your comment. About the staghorn stones and their association with the CKD, we added in the discussion:

“The increase in prevalence of CKD in our population may be due to the coexistence of several clinical associations that accelerate renal function damage. Although T2DM was a statistically significant factor in the adjusted analysis by BMI, sex, and age, it was more frequent in patients without CKD, suggesting that T2DM is not an exclusive risk factor for CKD.”

Also, we added:

“Previous studies have been observed an association between staghorn stones and CKD [19]. Interestingly, this study did not show a significant difference between groups in staghorn stones frequency, which suggests that recurrence of NL and surgical events are independent risk factors for CKD. Remarkably, the stones sizes and composition (HU) in both groups were similar.”

In addition, was added the CKD frequency in patients with ≤50 years old and >2 surgical events (Table 2): “The CKD frequency in patients aged ≤50 years old and >2 surgical events were significantly higher compared to those who reported fewer surgical events (p<0.001)”, these results confirm that the number of the surgical events are associated with CKD.

There are other issues that all are related to the methodology. For example, as shown in table 1, patients were not matched regarding age and hypertension. Renal function status before intervention is not clear. If we want to relay on the finding, we should conclude that type2DM is associated with lower risk of CKD (table 2) in patients with kidney stone!

Response 3: Your suggestions were added in the limitations section. Due to all patients were referred from other medical units it was not possible to match by age and hypertension. Also, it was complicated to assess the renal function before each surgical event from the patients, especially with their recurrence of NL.

Regarding the T2DM, we added:

“Although T2DM was a statistically significant factor in the adjusted analysis by BMI, sex, and age, it was more frequent in patients without CKD, suggesting that T2DM is not an exclusive risk factor for CKD.”

Reviewer 3 Report

The manuscript can be accepted after considering the following comments. The manuscript title needs to be more specific. There are many typos in the submitted manuscript. Check!!!! The abstract should be started with broad ideas, problems,and the suggested proposal, and finally, the obtained data. The keywords are very public, please specify them without repetation for the title words It is more preferable to introduce the data of Table 1 in 3 figures.

Author Response

Thank you for taking the time to review this manuscript and provide us with your comments and suggestions. Please, find down below the answer to your comments and suggestions point by point.

The manuscript can be accepted after considering the following comments. The manuscript title needs to be more specific. There are many typos in the submitted manuscript. Check!!!!

Response 1: The title was changed to “Recurrence of nephrolithiasis and surgical events are associated with chronic kidney disease in adult patients”. In addition, the typos errors were corrected.

The abstract should be started with broad ideas, problems, and the suggested proposal, and finally, the obtained data. The keywords are very public, please specify them without repetation for the title words

Response 2: The abstract and the keywords were modified according to the reviewer's suggestions.

“Abstract: Nephrolithiasis (NL) is a public health problem in the Southeast Mexico population by its high prevalence and recurrence. The evolution of this pathology can result in renal damage and may even cause chronic kidney disease (CKD), leading to a reduced glomerular filtration rate (GFR), decreased kidney function, and kidney loss in advanced stages. However, few studies support this evidence in the population. Thus, the present study aims to determine risk factors associated with CKD in adult patients in an endemic population of Mexico. A case-control study was carried out with patients diagnosed with NL. Additionally, the clinical information of patients (age, weight, height, blood pressure, comorbidities, and time of progress of NL), characteristics of the stones (number, location, and Hounsfield units), and biochemical parameters were collected. The results showed that recurrence of NL was associated with CKD (OR 1.91; 95%CI 1.37–2.27; p=0.003). In addition, male sex (p=0.016), surgical history (p=0.011), bilateral kidney stones (p<0.001), and urinary tract infections (p=0.004) were other factors associated with CKD. Interestingly, thirty-two patients younger than 50 years old with >2 surgical events presented a significant decrease in GFR (p<0.001).”

 “Keywords: Renal lithiasis; endemic disease; risk factors; kidney injury.”

It is more preferable to introduce the data of Table 1 in 3 figures.

Response 3: We include the Figure 1 instead Table 1 as suggested.

“Figure 1. General characteristics of patients with nephrolithiasis according to GFR classification, presence, or absence of CKD. A. Age; B. Body mass index (BMI); C. Systolic and diastolic arterial pressure.”

Round 2

Reviewer 1 Report

Complete with current edits. Manuscript acceptable for publication

Author Response

Thank you for your comments and your time, we made all suggested changes.

Reviewer 2 Report

This is a well written manuscript; however, since the study is methodologically flawed, I think it is not appropriate for publication.

Author Response

(The authors gave the same response as above.)
